# Equine Influenza Virus: An Old Known Enemy in the Americas

**DOI:** 10.3390/vaccines10101718

**Published:** 2022-10-14

**Authors:** Juliana Gonzalez-Obando, Jorge Eduardo Forero, Angélica M Zuluaga-Cabrera, Julián Ruiz-Saenz

**Affiliations:** 1Grupo de Investigación en Ciencias Animales—GRICA, Facultad de Medicina Veterinaria y Zootecnia, Universidad Cooperativa de Colombia, Bucaramanga 680002, Colombia; 2Grupo de Investigación en Microbiología Veterinaria, Escuela de Microbiología, Universidad de Antioquia, Medellín 050010, Colombia; 3Facultad de Medicina Veterinaria y Zootecnia, Fundación Universitaria Autónoma de las Américas, Circular 73 N°35-04, Medellín 050010, Colombia

**Keywords:** equine influenza, vaccines, H3N8, America, zoonosis

## Abstract

Equine influenza is a highly contagious disease caused by the H3N8 equine influenza virus (EIV), which is endemically distributed throughout the world. It infects equids, and interspecies transmission to dogs has been reported. The H3N8 Florida lineage, which is divided into clades 1 and 2, is the most representative lineage in the Americas. The EIV infects the respiratory system, affecting the ciliated epithelial cells and preventing the elimination of foreign bodies and substances. Certain factors related to the disease, such as an outdated vaccination plan, age, training, and close contact with other animals, favor the presentation of equine influenza. This review focuses on the molecular, pathophysiological, and epidemiological characteristics of EIV in the Americas to present updated information to achieve prevention and control of the virus. We also discuss the need for monitoring the disease, the use of vaccines, and the appropriate application of those biologicals, among other biosecurity measures that are important for the control of the virus.

## 1. Introduction

Equine influenza is a highly contagious disease distributed worldwide that infects equids [1,2]. New Zealand, Australia, and Iceland are the only territories considered free of the diseases [3,4,5]. The equine influenza virus (EIV) belongs to the *Orthomyxoviridae* family with a segmented negative RNA genome, which can be subtyped based on two glycoproteins: hemagglutinin (HA) and neuraminidase (NA) [1,3].

This virus, with a diameter of 80–120 nm, is considered one of the most important respiratory viral pathogens in horses. It has been established that EIV evolved from a common ancestral progenitor in the influenza A virus (IAV), with a natural reservoir in aquatic birds sharing the same host cell surface receptors as avian influenza viruses [6,7]. Two different avian-origin interspecies transmission events originated the noncirculating H7N7 subtype in 1956 and H3N8 in 1963 [8,9]. H3N8 is the primary EIV subtype, and this subtype has been divided into two lineages since the 1980s: the American and Eurasian lineages. The American lineage is divided into the Kentucky, South America, and Florida sublineages, whereas the latter is divided into two subtypes: Florida Clade 1 (FC1) and Florida Clade 2 (FC2) [10]. More than 85 outbreaks in Africa, Asia, Europe, and Australia have been attributed to FC1; however, FC2 is predominant in America and Asia [3].

## 2. Virus Classification

The *Orthomyxoviridae* family is divided into the genera Alphainfluenzavirus, Betainfluenzavirus, Gammainfluenzavirus, and Deltainfluenzavirus, which are influenza A, B, C, and D, respectively, and are classified according to the antigenic differences in the NP (nucleoprotein of the nucleocapsid) and in the M (matrix protein) [11]. Influenza A subtypes have been classified according to the hemagglutinin (HA-18 subtypes) and neuraminidase (NA-11 subtypes) [12]. Type A viruses primarily infect horses, swine, and humans [13], and the emerging influenza D virus (IDV) can also affect farm animals, including horses, bovines, and swine [14,15]. Recently, in equine populations from the Midwestern United States, the cocirculation of two lineages of IDV was reported, supporting the need for further surveillance of IDV viruses in agricultural species [16].

Influenza A viruses are composed of eight 13.6 KB negative polarity segments [17]. There are ~500 projections in the envelope of the virus, 80% of which correspond to the hemagglutinin protein and the remaining 20% correspond to the neuraminidase protein [10,18] (Figure 1). The proteins encoded by the segmented genome include structural proteins (called HA and NA), nucleoprotein (NP), matrix proteins (M1 and M2), three polymerase proteins (PB1, PB2, and PA), one nuclear export protein (NEP) that is also called nonstructural NS2, and one nonstructural protein called NS1 [11]. HA is an envelope antigenic glycoprotein that can bind to red blood cells, leading to agglutination. It promotes the binding of the virion to cell surface receptors (neuraminic and sialic acid). NA is an envelope protein whose enzymatic activity results in the liquid formation of mucus, thus contributing to the spread of the virus [13,18].

The H3N8 subtype was isolated in Florida (USA) in 1963 [19]. A phylogenetic analysis of HA determined that H3N8 evolved as a single lineage until the mid-1980s and then diverged into two different evolutionary lineages: an American lineage and a European lineage. Subsequently, the American lineage became a sublineage of South America, Kentucky, and Florida. The Florida sublineage is currently predominant and has diverged into FC1 and FC2 [20]. Currently, the H7N7 subtype is considered to be no longer circulating or extinct in horses, and no known circulation of this subtype has been recorded since the end of the 1970s [21].

## 3. Replication EIV

After inhaling the virus, most viral particles are captured by the mucosa of the respiratory system. EIV primarily damages the lower respiratory tract of ciliated epithelial cells, resulting in the inability to remove foreign bodies and substances. Early changes caused by the influenza virus in the epithelium of the upper airway are variable, including necrosis followed by desquamation of these cells into the luminal space; necrotic cells may also undergo phagocytosis by macrophages [22].

The HA glycoprotein binds to sialic acid receptors located on the surface of the host cell and subsequently generates the endocytosis of the viral particles, thus leading to an acidic pH [1]. The infection cycle begins when the viral particle binds to the sialic acids of the cell membrane. Endosomal acidification results in a three-dimensional modification of HA, which is crucial for the fusion of viral proteins with the endosomal membranes. Subsequently, the proton channels of the virion (M2) activate because of the dissociation of the vRNP complexes of the M1 matrix proteins that provide stability to the viral architecture by interacting with the viral envelope and nucleocapsid. VRNPs that are released into the cytoplasm are translocated to the nucleus for their transcription and replication. For this purpose, the viral genome uses the import-specific α/β nuclear import system that requires the presence of a nuclear localization signal. Furthermore, NP, M1, and the nuclear export protein (NEP) are also introduced into the nucleus of the host cell [17]. From the viral polymerase complex PB2, PB1, and PA, first, the positive sense copy is synthesized, that is, cRNA, from which copies of the viral RNA are produced. This activity begins when PB2 snatches the 5-cap structure of the host cell’s messenger RNAs, and this transcription stage continues until the polymerase complex is stopped in a polyadenylation signal [23] (Figure 2).

NEP (14 kDa, 121 aa) directly binds to the Chromosome Region Maintenance 1 (CRM1) export cells. NEP binds to the M1 protein using its C-terminal domain. Multiple studies have demonstrated that NEP assimilates an adapter between vRNPs and the nuclear export system via a specific signal (NES) [24,25]. NESs are binding sequences rich in hydrophobic amino acids, particularly leucine. The N-terminal domain of NEP mediates the dependent binding of RanGTP to CRM1 and translocates to the cytoplasm through the nucleoporins of the nuclear pore complex (NPC). The hydrolysis of RanGTP to RanGDP leads to the dissociation of the export complex and release of the vRNP viral load in the cytoplasm of the host cell. The virus is later released from the infected cells through budding. This viral replication lasts 6 h [17].

The assembly process of the eight segments is mediated by the M1 and M2 proteins; the latter mediates the incorporation of the vRNPs into the assembled particle, where HA and NA associate with the lipid rafts for subsequent packaging together with the other proteins [1,23].

## 4. Pathogenesis

Serum amyloid protein acute-phase proteins may be reported after infection by EIV, which can appear as an inflammatory marker, with an increase in concentrations during the first 48 h. However, these values return to baseline levels at days 11 and 22 post infection. The concentrations of this protein are positively correlated with the severity of clinical signs and an increase in body temperature [26]. Serum amyloid increases during influenza infection in animals and interacts with macrophages and phagocytes, inhibiting virus replication [27].

In the necropsy results, morphological changes were limited to the lungs and lymph nodes, primarily showing hyperemic nasal mucosa. Swollen retropharyngeal and pulmonary lymph nodes are observed, as well as petechial hemorrhages and mucopurulent exudate in the bronchi accompanied by pulmonary edema [28].

### 4.1. Clinical

Elevated body temperature between 39 and 41 °C, a dry cough, and rapid transmission of infection determine the presumptive clinical diagnosis of equine influenza. Severely infected horses may present with inspiratory and expiratory wheezing and crepitations. Other signs, such as anorexia, lethargy, and nasal discharge, may be present; the latter is initially serous with a tendency toward becoming mucopurulent within 1–2 days, resulting from a secondary bacterial infection. One or more of these clinical signs could not be present in vaccinated animals [29,30]. Affected animals recovered within 7–10 days. Similar to vaccination response, animals are normally protected against a new infection for approximately one year [6,31]. Morbidity can reach up to 100% in nonimmunized populations [32].

It takes at least three weeks for the respiratory epithelium to regenerate; therefore, opportunistic bacteria can invade easily, and specific antigens can penetrate more deeply, leading to conditions such as bacterial bronchopneumonia, allergic bronchitis, and bronchiolitis [33]. Inflammation in the airways is characterized by peribronchial and peribronchiolar hyperemia with the infiltration of mononuclear cells, exudation of neutrophils and macrophages in the airways and, in certain cases, interstitial pneumonia together with congestion and alveolar edema. Myocarditis has been reported in certain cases [1,26].

### 4.2. Molecular

Cells infected with the EIV undergo apoptosis [34]. IAV viruses can generate apoptosis in the tissue by the intrinsic or mitochondria-dependent pathway and the extrinsic pathway induced by ligands that bind to receptors, subsequently leading to cell death. There is also a secondary necrosis process in affected cells, which is presented by an N-terminal fragment that induces gasdermin E/DFNA5, which is cleaved by a caspase. Necroapoptosis regulated by RIPK1/3 and MLKL proteins can also occur; these three processes compromise the integrity of the epithelial cell barrier [35].

The immune response in equines to the EIV is initially evidenced by the presence of immunoglobulins, followed by the innate cellular response, which is activated by pathogen-associated molecular patterns (PAMPS), detected by toll-like receptors 3, 7, and 8; NK cells; neutrophils; and monocytes. The adaptive response is associated with the activity of CD4+ and CD8+ cells [36].

## 5. Epidemiology

The virus is highly contagious and is transmitted between infectious and susceptible individuals, principally via the respiratory route; therefore, it is considered a mandatory notification sickness by the World Organization for Animal Health (WOAH, formerly the OIE) [37]. Indirect transmission can occur through fomites, particularly via contaminated vehicles [3]. Fomites may also include clothes of the personnel working with the equines; for this reason, it is important to maintain the control and hygiene measures in the herds, such as the disinfection of hands and clothing, as well as restrict access to external personnel during an outbreak [38,39].

Cases of EIV have been reported in different countries in America (Table 1). The presence of H3N8 FC1 was reported in 2018 in Chile [40]. The H3N8 subtype was reported and identified in Brazil [41,42]. The H3N8 subtype FC1 has been reported in Argentina, and H3N8 has been reported in Uruguay [43]. The H3N8 subtype has been identified in the United States and Canada [1]. From a phylogenetic point of view, group 1 of the H3N8 subtype, which had ancestry with one of the first H3N8 strains from Sao Paulo, was reported through a phylodynamic analysis in Brazil [42]. Moreover, Group VII belongs to the H3N8 subtype, which mostly comprises Argentine strains reported in 1985. South American Clade 1 was composed of Argentine strains, and South American Clade 2 was composed of Argentine and Chilean strains. Florida Clade 1 was composed of strains reported in Chile, Argentina, and Uruguay, which targeted the equine population between 2011 and 2012. They reported substitutions in HA antigenic site B in the amino acid comparison analysis, and this result can compromise the antigenicity of the virus [44].

The first outbreak in a human population exposed to horses was recorded in 1957 in Kharkov (Ukraine). This diagnosis was confirmed by serology [48,49]. Experimental investigations carried out in the 1960s in the United States demonstrated the zoonotic impact of the EIV. In these studies, the equines presented clinical signs, such as fever, sore throat, and a runny nose [50]. Later, in Mongolia, where there is a recognized equine culture, serological studies have been reported with a positivity of 0.091% in the human population related to equines [51,52]. The presence of antibodies in humans linked to equine populations has also been reported in the United States (11%) [2] and Australia (0.09%) [53] in individuals who live with horses. In Chile, an equine influenza infection was reported in a veterinary medicine student who was caring for symptomatic animals [54]. Differences between EIV and human influenza virus sialic acid receptors may act as a restriction factor for the interspecies transmission from equine to humans and for the efficient human-to-human spread of EIV [55]. However, in addition to sialic acid, other viral and host genes could be responsible for influencing host range and interspecies jumps that still need to be clarified [56].

The equine influenza virus H3N8 presented another jump in the species barrier and was established in the dog population in 2004 as the cause of disease among racing greyhounds, and the virus was transferred within a year to racing tracks in many places, including the United States, Australia, and the United Kingdom. The canine influenza virus H3N8 was clinically characterized in 2005, for which a hyperacute syndrome and a mild one could be identified; later, the RNA sequence of A/canine/Florida/43/2004 was analyzed, in which they found a similarity of 96% when compared with the circulating equine strains of the time [57]. This similarity may be a consequence of the close relationship that the canine and equine populations have had on racetracks for decades. In September 2002, an outbreak of severe respiratory disease in English foxhounds in the United Kingdom was caused by EIV; however, the mechanism of transmission remains unclear, although direct respiratory transmission from horses to dogs was suspected [58]. On the other hand, in the 2007 Australian outbreak of canine influenza, transmission was linked to cases of equine influenza in racing horses [59,60,61].

## 6. Molecular Epidemiology

Phylogenetic analysis of the EIV HA gene demonstrated that H3N8 EIV evolved as a single lineage for at least two decades and diverged during the mid-1980s into the American and European lineages, which are named as per their geographic origin. The latter has been included in the Eurasian lineage, with reports being made from 1989 to 1994 [62]. Strains within the American lineage diverged into the South American, Kentucky, and Florida sublineages. Currently, the Florida sublineage is predominant and has evolved into two antigenically different clades, FC1 and FC2, which have been mostly isolated in North America. Two other clades have been reported in South America: South American Clades 1 and 2 [1] (Figure 3).

## 7. EIV Risk Factors in the Americas

According to the different reported papers in the Americas, different risk factors have been highlighted as remarkably associated with EIV infection and transmission within and between herds.

**Age**. Younger animals used to have a lower level of EIV antibodies than older individuals, increasing their susceptibility to infection. In Brazil, Daly et al. reported a statistically significant difference (*p* = 0.001) for the presence of antibodies between horses under 5 years of age and horses between 5 and 14 years of age, who had higher levels of antibodies [20]. In the United States between 2010 and 2013, it was reported that the mean age of horses positive for EIV was 4.7 years: in the multivariate analysis, being between 1 and 5 years was a factor associated with the positive diagnosis for equine influenza with an OR (7.37) and a *p* value of 0.001, and being between 6 and 10 years old was a factor associated with an OR (8.94) and a *p* value of 0.001 [50]. In the equine influenza outbreak that occurred in 2012 in Argentina and Uruguay, the most affected animals were those between two and three years of age [44]. Additionally, in the 2018 outbreak in Argentina, the most affected individuals were between 1 and 6 years of age [47]. In Mexico, between 2010 and 2011, a greater presence of antibodies against influenza H3N8 was found in individuals older than 2 years, which indicates that young individuals were more susceptible to the presentation of the disease [63]. Although every age group appeared to be susceptible to EIV, age-dependent susceptibility may be an important feature for epidemiological control and may be related to increased contact and risk with other infected animals [30,64].

In foals, the onset of the adaptive immune response is delayed in comparison to the adult horse. The foal’s adaptive immune response seems to be immature for adaptive immune parameters, such as immunoglobin antibody production and adaptive T-cell responses [65]. Moreover, oxidative stress has been observed in young growing horses that have just started exercise training [66]. These findings could explain, at least in part, the increased susceptibility of young horses to infectious agents such as EIV and the reduced responsiveness of foals to the EIV vaccine. Colostrum maternal antibodies from EIV-vaccinated mares induce protection in the foal [6]; therefore, it is recommended to vaccinate pregnant mares four to six weeks before delivery, which prevents the presentation of severe equine influenza clinical signs in foals [6].

**Equine movement and travel**. National and international increases in horse movement, especially competition horses, have been related to the transmission and spread of EIV outbreaks and could potentially lead to an increased global spread of infectious equine diseases [67]. In Mexico in 2012, a higher presence of antibodies against equine influenza H3N8 was reported in those horses linked to sports activities such as horse riding (OR = 1.36) and jumping (OR = 1.32), which traveled more due to attendance at sporting events [63]. The relationship between the appearance of influenza outbreaks and the displacement of competing individuals was also reported in Argentina [44].

The 2012 EIV outbreak throughout South America was first reported in Chile in December 2011 and spread rapidly through Chile, Brazil, Uruguay, and Argentina [44,68]. Phylogenetic analysis of the EIV from different countries showed that South American isolates from 2012 were closely related to the 2011–12 isolates from the USA, suggesting that these viruses are likely to have originated from the USA and with a possible importation to Chile and a subsequent spread via travel through South America [68]. Additionally, in 2012, a group of endurance horses were transported from Uruguay to Dubai carrying EIV, highlighting the risk of EIV spread to multiple countries via international travel [69].

In Chile, in the 2018 outbreak, most of the EIV-positive horses were competitive, either from polo, jumping, or racing sports. The outbreak rapidly spread across Chile, facilitated by the rodeo season, which increases horse movement [40]. The analyzed phylogenetic relationship with international sequences showed that this new introduction of EIV to South America was related to concurrent outbreaks occurring globally in Europe, Asia, and North America and introduced to the country as a result of international horse movements [40], as has been previously reported for EIV and other equine infectious diseases [67].

**Vaccination status**. Despite the availability of EIV vaccines, multiple challenges remain to achieving effective immunization in horses and control of EIV, including the evolution of the virus, vaccine breakdown, vaccination-induced short-lived immunity, the inability of vaccines to induce sterilizing immunity, and, in young horses, interference with maternally derived immunity [6]. Antigenic differences between field strains and vaccine strains may affect vaccine efficacy due to a low specific immune response [70], with low to mild cases even in vaccinated animals but reducing severe disease. In Latin America, the use of outdated vaccine strains, incomplete EIV vaccination programs, or updated vaccines without sufficient antibody production may be a contributing factor to the presence of the disease [40,47,68].

The first outbreak in South America in which the presence of cases due to an ineffective vaccination program was evident was in Chile in 1963 with the introduction of H3N8 EIV strains into a population [44]. In 2015 in Brazil, 37.5% of infected horses were vaccinated with outdated strains (A/equine/Kentucky/97, A/equine/SouthAfrica/4/2003, A/equine/Kentucky/94) [42]. For the 2018 Argentinean outbreak, it was reported that 76% of affected horses had been vaccinated with outdated A/eq/Kentucky/1997 strains [1,47]. The situation was similar in Uruguay in 2018, for which the authors mention that many of the vaccines distributed in this country were outdated, associating them with the presentation of the outbreak [43].

Although infection has been reported in vaccinated animals, it is clear that most of the cases during an outbreak belong to EIV-unvaccinated animals [6,70]. As the rate of change in HA/NA leading to antigenic drift is slower in EIV than in human influenza viruses, it is accepted that relatively “old” vaccine strains could protect horses longer than is known for humans. However, those strains will be antigenically no longer representative of EIV circulating strains and will become obsolete as vaccine strains sooner or later [71,72]. In addition, EIV vaccine efficacy also depends on the vaccine adjuvant, which is included to stimulate the equine immune response to the target antigen either in the whole inactivated or subunit virus vaccines and modified viral vector vaccines [70].

**Low herd immunity**. To date, vaccination remains one of the most efficient methods of prevention against several major equine infectious diseases; however, the establishment of efficacious and long-lasting protective immunity at the herd level is a complex issue. Herd immunity reduces the size and frequency of epizooties at the population level and reduces virus shedding; however, sterilizing immunity is rarely observed, and vaccine breakdown may possibly occur [73]. High levels of vaccine coverage (80–90%) have been reported to be required to provide herd immunity against infection [73,74].

For the 2018 Chilean outbreak, it was concluded that one of the causes associated with the spread of EIV in the country was the low vaccine coverage, especially in thoroughbreds. In Chile, equine vaccination is not mandatory, and in the country, there is only an average of 58,000 doses available per year, that is, enough doses for 20% of the total equine population being too low to achieve a constant herd level of protection [75]. In Mexico, in 2015, 114 equine samples with clinical signs associated with influenza were investigated, and of those, 75% were positive for equine influenza: none of the equines were vaccinated against equine influenza [76]. Even in the United States, 85% of horses positive for H3N8 EIV (60/761) had no or undetermined vaccination status [50]. In Canada, in a study conducted on 23 respiratory outbreaks that occurred between 2003 and 2005, it was reported that only 36% of horses were vaccinated against equine influenza [45]. These results highlight the fact that there is not enough vaccine coverage in the Americas to avoid or prevent a possible vaccine breakthrough. Additionally, achieving a protective herd immunity level should be the most important factor associated with the presentation of EIV outbreaks in the Americas.

Some of these risk factors have been extensively reported for IAV infections in humans, which are mainly related to international travel and outbreaks appearing through contact with fomites or microdroplets from infected individuals [56].

## 8. Studies on EIV Occurrence and Reported Outbreaks

**Argentina**. Different EIV subtypes have been reported in the country. In 2012, an extensive outbreak was reported in the country’s equine population, which probably occurred due to the movement of equines between Uruguay and Argentina [44]. In 2016, a high frequency in several cities in Argentina during an outbreak reported 67.5% of the H3N8 subtype through molecular diagnosis. In 2018, a molecular prevalence of 58% was reported for the FC1 subtype H3N8 from nasopharyngeal samples taken, and a seroprevalence of 70% was determined via hemagglutination inhibition testing against A/eq/Argentina/E-2345- 1/2012 (FC1 strain) of horses symptomatic for influenza [46,47].

**Brazil**. The first Brazilian outbreak caused by the H3N8 subtype (A/Equi/SP/63) occurred in 1963. New outbreaks caused by H3N8 have been described in 1969, 1985, and 1988 in Sao Paulo. Multiple investigations have been reported in which the presence of antibodies was evaluated; one of these outbreaks was in 2012, and was caused by H3N8 A/equine/Rio Grande do Sul/2012, which had a high impact on the equine population in both vaccinated and nonvaccinated animals [41,42,68]. Favaro et al. analyzed 83 equine sera in 2017, of which 100% were seropositive for H3N8. These samples were taken from healthy unvaccinated horses [41].

**Canada**. EIV has been reported since 1979, when 63.8% of horses in racetracks from western Canada with respiratory signs showed the presence of influenza A antibodies [77]. Outbreaks caused by H3N8 were reported in 1991 and 1992. Furthermore, they reported a 76% seroprevalence for the H3N8 subtype in race horses with respiratory clinical signs in 2000 [78]. In 2010, Díaz et al. reported a seroprevalence of the H3N8 subtype in 24.4% of horses with respiratory clinical signs [45].

**Chile**. Multiple outbreaks have been recorded throughout the country. From 1963 until 2018, EIV was reported in 1963, 1977, 1985, 1992, 2006, 2012, and 2018 [75,79]. The 1963 outbreak occurred almost simultaneously with the one in the United States [19]. Moreira et al. reported a frequency of infection by the H3N8 subtype of 84% during the outbreak of 2018 through the molecular diagnosis of sick animals with clinical signs of influenza, including nasal discharge, fever, and cough [75].

**Colombia**. Equine influenza was reported in 1982 for the first time in 41 symptomatic horses from five stables in the Bogotá Savannah during an outbreak that affected >3000 horses. The diagnosis was confirmed by the significant increase in three viral agents identified as influenza type A/Equi2/Miami/H3N8. After this, outbreaks were reported in 1998, 2005, 2010, and 2018; however, the subtypes involved in these outbreaks have not been identified. A frequency of infection of 4% for influenza was reported among 220 healthy horses in 2013 in the eastern areas [80]. A rapidly spreading outbreak was reported in 2018, which was related to equestrian sporting events, and cases were reported in 22 out of the 32 departments, distributed from the Midwest to the North and Southwest of the country.

**Mexico**. In 2008, 186 equines were sampled, of which 25% presented antibodies against H3N8 EIV, and 97% of these equines did not have clinical signs compatible with influenza [81]. Between 2010 and 2011, 212 horses without signs of equine influenza were sampled, of which 45.78% had antibodies against influenza H3N8 [63]. In 2019, Plata-Hipólito analyzed 114 equine serum samples, out of which 92.6% were positive. The horses included in the study presented clinical signs of equine influenza [76].

**United States**. It has been assumed that the first introduction of the EIV to North America and the first report was in 1963 (H3N8) came from horses imported from the south cone of the continent, mainly from Argentina [79], which was denoted A/Eq/Miami/1/63 [21]. Currently, Florida Clade 1 is endemic in the United States, and the reports associated with Clade 2 belong to imported horses [82]. A prevalence of 9.3% was reported in horses with acute respiratory clinical signs in a study conducted from 2010 to 2013 [50], and recently, a rt-qPCR frequency of 6,8% was reported in a voluntary sampling of 10,296 equids with acute onset of fever and/or respiratory signs for rostral nasal secretions in the country between 2008 and 2021 [83]. Phylogenetic analyses of EIV H3N8 strains from 2012 to 2017 based on the HA gene have shown that H3N8 strains from the US underwent antigenic drift and fixed multiple substitutions accumulated in the HA gene [84] and that the H3N8 FC1 North American lineage was the major source of globally dominant EIV variants and spread to other global regions [85].

**Uruguay**. The first laboratory diagnosis of EIV in Uruguay occurred in 1963 [86]. Since then, it has been periodically presented and has been notified annually to the World Organization for Animal Health since 2006 [43]. During the 2012 outbreak, approximately 2500 horses were affected, and some of these equines were subsequently transported to Dubai, putting the Asian equine population at risk [46,69].

**The 2018 outbreak in Latin America**. In 2018, an intercontinental EIV was reported from Argentina to Colombia. The EIV was confirmed in Chile in January 2018, and outbreaks were reported in Argentina and Uruguay in March and June of the same year. The strains reported and identified in the outbreak were A/equine/Concepcion/RO7C/18; A/equine/Argentina/E-434-6/2018; and A/equine/Colina/Chile_15373/2018, which belong to Florida Clade 1. As shown in Figure 3, all reported sequences in databases confirm the monophyletic origin of the intercontinental outbreak.

In Argentina, it was determined that 24% of affected animals reported vaccination in the past months, thus indicating a failure in the vaccination scheme, which included the strain recommended by the OIE that, to date, contains strains of Florida sublineages 1 and 2 A/eq/Argentina/E2345-1/2012 and A/eq/Meath/2007, respectively. However, the clinical signs of these vaccinated horses were mild [47]. In Chile, the clinical signs of affected horses were moderate to mild, perhaps due to the presence of vaccination with “old” or outdated vaccine strains, as discussed before, lowering the severity of disease [71]. Additionally, more complex and severe cases were reported, which were associated with secondary infection by *Streptococcus equi* [40]. In Uruguay, the outbreak caused by the equine influenza virus had a duration of one month, for which cases were reported in eight departments of Uruguay [43]. In other countries, such as Colombia and Peru, the EIV was confirmed by RT–qPCR, but only WAHIS confirmation is available. Certain causes associated with the outbreak were low vaccination coverage, low herd immunity, and equestrian events between localities and countries.

## 9. Prevention and Control

Isolating symptomatic animals and monitoring their body temperature twice a day is recommended for the prevention and control of equine influenza. Furthermore, because outbreaks are associated with the introduction of new horses to the facility, animals that have just entered a population should be quarantined for 21 days and vaccinated [87].

It has been shown that EIV infects high-performance horses [88]. Arduous exercise over a prolonged period of time may result in low-level airway inflammation, tissue remodeling, cellular stress, and immunological alterations, among other alterations, allowing the entry of microorganisms that subsequently affect the respiratory system [89]. Furthermore, horses exposed to other horses, as is common in racing events, are more vulnerable to the virus [64]. In addition to physical contact between animals, travel conditions, mixing with other vaccinated and unvaccinated horses, and poor biosecurity could contribute to an increased risk of infection. Long- and short-term transportation affects physiological, endocrine, and immune responses as soon as 15 min post-transport, implying that horses may be vulnerable to disease during and almost immediately after short-term transport [90]. In mathematical modeling, mixing vaccinated and unvaccinated animals and the use of isolation could not only reduce disease burden in equine populations but also reduce disease transmission and decrease the cumulative incidence of EIV [91,92]. In 174 thoroughbred foals, the antibody response to primary EIV vaccination was assessed, finding that there is a positive relationship between vaccination doses and EIV antibody titers, which demonstrates the poor response to primary vaccination, in addition to showing the relevance of serological surveillance to evaluate herd immunity and specific EIV antibodies in foals [93].

The protection generated by the vaccine is attributed to the induction of antibodies against the glycoproteins of the viral surface, particularly HA. Live-attenuated vaccines and inactivated vaccines are vaccines that have shown better protection to date. The latter is recommended by the WOAH because it includes the representative viruses of both Clades 1 and 2 of the Florida sublineage [69]. In addition, the WOAH Expert Surveillance Panel on Equine Influenza Vaccine Composition recommends updating the vaccines that are sold; however, it is difficult due to the time and cost that this process implies for the commercial market [56]. Currently, other types of vaccines have been developed, such as adjuvanted subunits and genomic vaccines [6,56]. In the development of new vaccines, it is important to keep in mind the type of adjuvant employed because it generates a better immune response, even similar to the immune response developed by a new field strain, in addition to generating a rapid decrease in clinical signs [70]. It has been stated that in addition to decreasing the severity of disease and reducing the morbidity rate during outbreaks, vaccination ensures immunity for the animals, improves recovery, and reduces the economic impact of this disease [94,95].

Updating the vaccine strain is as important as understanding the differences between the composition of these vaccines and the mechanism by which they induce immunity, causing differences in vaccine efficacy [70]. Multiple EIV vaccines are available commercially and can be classified into three groups based on vaccine type: 1―adjuvanted, whole or subunit, inactivated virus vaccines; 2―adjuvanted, modified viral vector vaccines; and 3―live attenuated virus vaccines [95]. To date, only live attenuated virus vaccines can induce long-lasting protective immunity by activating multiple components of the immune system [6].

The vaccination schedule for inactivated and recombinant vaccines begins at five to six months of life, with two doses with an interval of 4 months between them. A booster must be applied six months after the initial vaccination program. This booster provides protection for 12 months [96,97]. However, it has been reported that annual booster vaccination should not be relied on as the sole preventative measure against EIV, and increasing the frequency of booster vaccinations may be beneficial, particularly in young horses in places with high EIV transmission [88].

In addition to vaccination, other important actions could help to control and prevent current and future EIV outbreaks. Sanitary and biosafety education for horse owners and managers helps to obtain preparedness for an EIV outbreak and a greater interest in information about infection control [98]. The implementation of biosafety practices, including reduced movement of horses and people in and out of the facility, the use of footbaths in the facilities, and control of the virus in maternal antibodies, reduces EIV transmission efficiently and reduces the risk of new outbreaks [38,99]. It is also essential to quarantine, especially those individuals who enter the paddocks or herds for a minimum of two weeks [38,47]. Serological tests of individuals who have just entered the property, separating traveling animals from the equine population that permanently resides on the premises, maintaining limits between the different groups of equines, and incorporating personal hygiene measures could also help to control an outbreak [30,99,100].

## 10. Conclusions

The equine influenza virus is primarily caused by the H3N8 subtype, both in the Americas and worldwide. It is one of the equine diseases with greater clinical and economic implications; the latter is associated with the cancellation of training, veterinary treatments, and equestrian events [46]. Based on the phylogeny, viruses associated with outbreaks are different in America from those that are currently in commercial vaccines, which is why these outbreaks reported in Latin America have affected both vaccinated and unvaccinated populations—particularly competitive horses.

Finally, it is important to monitor the EIV in animals coming from other countries or continents to control it before it enters the new equine population. Prevention and control measures, such as quarantine and the isolation of infected or suspected horses, should not be left, but the best strategy for the prevention of equine influenza is vaccination and annual booster vaccination or shorter booster periods in high EIV transmission settings.

## Figures and Tables

**Figure 1 vaccines-10-01718-f001:**
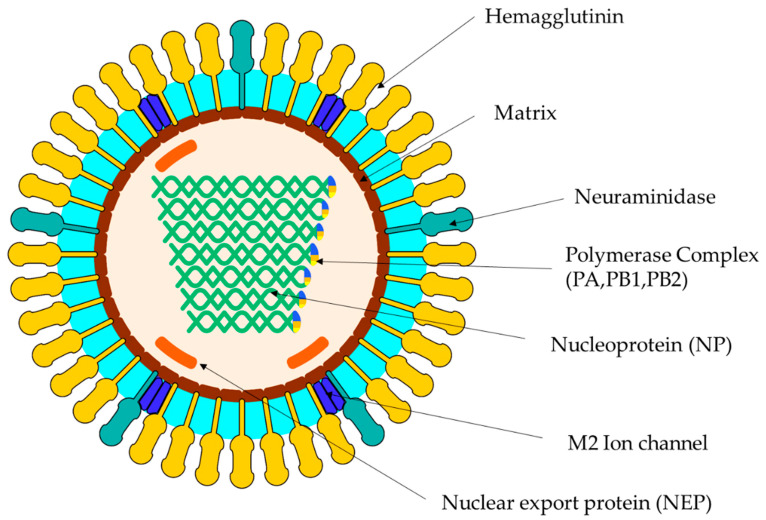
Diagram of the equine influenza virus structure and its genome. The diagram of the virion shows multiple proteins. See text for references.

**Figure 2 vaccines-10-01718-f002:**
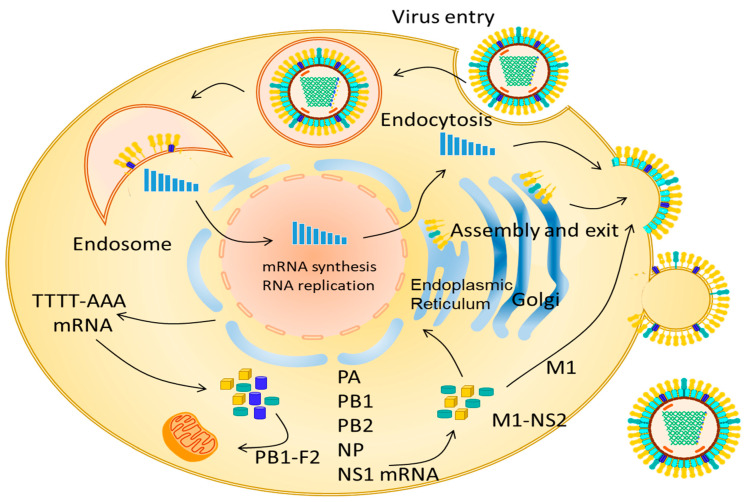
Schematic view of influenza A virus replication within an epithelial cell of the respiratory tract.

**Figure 3 vaccines-10-01718-f003:**
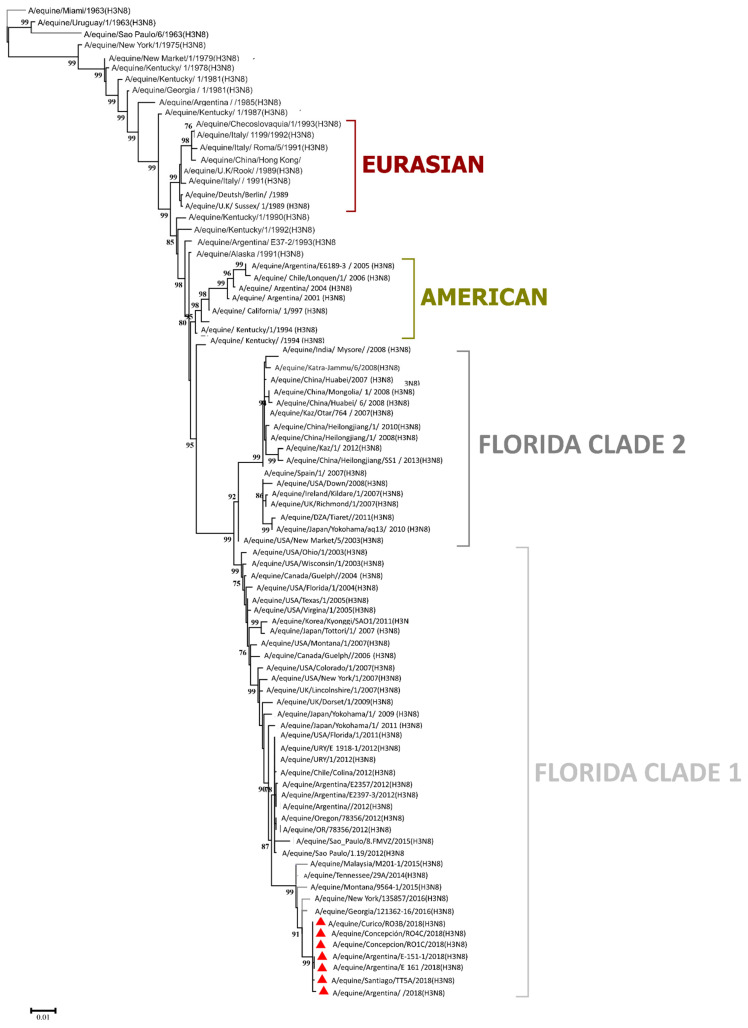
Maximum credibility for the complete HA gene of H3N8 EIV. Phylogenetic analysis of hemagglutinin (HA) gene nucleotide sequences from 86 equine influenza viruses (EIVs). The maximum likelihood tree was constructed using the stringent GTR + G algorithm, which was identified by using the best DNA/protein model tool available in MEGA 7. The reliability of the trees was assessed by bootstrapping with 1000 replications. Clades: Eurasian, American, Florida clade 1, and Florida clade 2. Red triangles denote samples belonging to the 2018 South American outbreak.

**Table 1 vaccines-10-01718-t001:** Characterization of EIV outbreaks in the Americas.

Country	Year	Subtype	Clade	Reference
USA	1963	H3N8	Predivergent	Sight, et al., 2018 [1]
Brazil	1980	H3N8	ND	Favaro, et al., 2017 [41]
Canada	2005	H3N8	ND	Diaz, et al., 2010 [45]
Uruguay	2012	H3N8	FC1	Perglione, et al., 2016 [46]
Argentina	2012	H3N8	FC1	Perglione, et al., 2016 [46]
Brazil	2015	H3N8	FC1	Favaro, et al., 2017 [41]
Argentina	2018	H3N8	FC1	Perglione, et al., 2020 [47]
Chile	2018	H3N8	FC1	Mena, et al., 2018 [40]

## Data Availability

Not applicable.

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
