# Peer review of "Equine Influenza Virus: An Old Known Enemy in the Americas"

_vaccines, 2022, doi:10.3390/vaccines10101718_

Round 1

Reviewer 1 Report

Among the respiratory diseases affecting the equine industry, equine influenza is among the leading pathogens. Monitoring disease dynamics and viral evolution is crucial to controlling the disease. Gonzalez-Obando et al. provide a review of the molecular, pathophysiological, and epidemiological characteristics of EIV with the aim of updating the knowledge gaps toward curbing the impact of the disease in the Americas. The review is in-depth, covering all aspects set out for investigation and timely, especially given the devastating episode of the 2018-2019 equine influenza outbreaks experienced in the region and across the globe.

The manuscript requires major structural adjustments before it can be deemed acceptable for publication. The author should consider restructuring the manuscript in chronological order as highlighted below:

1.      Introduction

2.      Virus classification

3.      Replication of EIV

4.      Pathogenesis

a.      Clinical

b.      Molecular

5.      EIV in animal models

6.      Epidemiology

a.      Molecular epidemiology

7.      EIV risk factors in the Americas

8.      EIV frequencies and reported outbreaks in the Americas

9.      The 2018 outbreaks in Latin America

10.  Prevention and control

11.  Conclusions

There are minor corrections the authors need to effect in the manuscript.

Abstract: line 18 – “among vaccinated and unvaccinated populations in America”. The authors need to identify the population that is being referred to here.

 Introduction: The authors should provide background on influenza A in birds and link it to how they diversify to equines.

Line 35-37: For consistency, use Florida clade 1 and 2 instead of “Florida 1 or 2 clade”.

Line 56: delete “acid” in front of neuraminic acid to avoid redundancy

Figure 1: the title should read “diagram of the …..” and not “draw”

Figure 1: change “matriz” to matrix

Figure 1: change “hemmaglutinin” to hemagglutinin

Figure 1: change “polymerasa” to polymerase

Figure 2: “ER” should be written in full.

Figure 2: change “golgy” to golgi

Line 186-192: the paragraph should be harmonized with the sub-section “EIV risk factors in the Americas”.

Line 232: change “frequencies” to “occurrence”.

 Line 233: For consistency with the use of abbreviation, use “EIV” instead of “IEV”.

Line 240: change “agglutination inhibition” to “hemagglutination inhibition”

Line 244-245: Sentence not in English language. Check and translate accordingly.

Line 249: What is IVD? Define at first mention.

Line 284: What do you mean by “Heq2Neq2”?

Line 321: “qPCR” should read RT-qPCR

Figure 3: change “clado” to clade

Conclusions: this should be expanded to provide in-depth discussion to reflect the vaccine types and vaccination schedule vis-à-vis the differences in the H3N8 viruses in circulation across the Americas, including what has to be done to be ahead of the virus as control measures.

Author Response

RESPONSE TO REVIEWERS COMMENTS - Submission Vaccines-1888753

Please find below our point by point responses to the comments regarding our Manuscript.

The changes are highlighted in Yellow in the file.

We would like to thank the Reviewers and Academic Editor for their helpful suggestions, for critical analysis of the manuscript, and for providing new discussion topics.

Reviewer 1

Among the respiratory diseases affecting the equine industry, equine influenza is among the leading pathogens. Monitoring disease dynamics and viral evolution is crucial to controlling the disease. Gonzalez-Obando et al. provide a review of the molecular, pathophysiological, and epidemiological characteristics of EIV with the aim of updating the knowledge gaps toward curbing the impact of the disease in the Americas. The review is in-depth, covering all aspects set out for investigation and timely, especially given the devastating episode of the 2018-2019 equine influenza outbreaks experienced in the region and across the globe.

The manuscript requires major structural adjustments before it can be deemed acceptable for publication. The author should consider restructuring the manuscript in chronological order as highlighted below:

  1. Introduction - 2.      Virus classification - 3.      Replication of EIV -
  2. Pathogenesis - a.      Clinical - b.      Molecular
  3. EIV in animal models - 6.      Epidemiology - a.      Molecular epidemiology
  4. EIV risk factors in the Americas - 8.      EIV frequencies and reported outbreaks in the Americas
  5. The 2018 outbreaks in Latin America - 10.  Prevention and control - 11.  Conclusions

R/. We agree to the reviewer. Text order suggestions were included

There are minor corrections the authors need to effect in the manuscript.

Abstract: line 18 – “among vaccinated and unvaccinated populations in America”. The authors need to identify the population that is being referred to here.

R/. We agree. it  was clarified in line 18

 Introduction: The authors should provide background on influenza A in birds and link it to how they diversify to equines.

R/. We agree. A sentence was Added.

Line 35-37: For consistency, use Florida clade 1 and 2 instead of “Florida 1 or 2 clade”.

R/. We agree. adjustment was made was included in the text in line 35 and 37

Line 56: delete “acid” in front of neuraminic acid to avoid redundancy

R/. We agree. the word “acid” was deleted

Figure 1: the title should read “diagram of the …..” and not “draw”

R/. We agree to the reviewer, the word “draw” was changed by “diagram”

Figure 1: change “matriz” to matrix

R/. We agree to the reviewer, the word “matriz” was changed by matrix

Figure 1: change “hemmaglutinin” to hemagglutinin

R/. We agree to the reviewer, the word “hemmaglutinin” was changed by hemagglutinin

Figure 1: change “polymerasa” to polymerase

R/. We agree to the reviewer, the word “polymerase” was changed by polimerase

Figure 2: “ER” should be written in full.

R/. We agree to the reviewer, the word endoplasmic reticulum was inserted in the diagram

Figure 2: change “golgy” to golgi

R/. We agree to the reviewer, the word golgi was to changed by golgy

Line 186-192: the paragraph should be harmonized with the sub-section “EIV risk factors in the Americas”.

R/. We agree to the reviewer. The information described in the paragraph was included in factors associated with the virus in the americas

Line 232: change “frequencies” to “occurrence”.

R/. We agree to the reviewer, the word “frecuencies” was changed by “occurrence”

 Line 233: For consistency with the use of abbreviation, use “EIV” instead of “IEV”.

R/. We agree to the reviewer. We changed the abbreviation in entire document.

Line 240: change “agglutination inhibition” to “hemagglutination inhibition”

R/. We agree to the reviewer, we changed “agglutination inhibition” by hemagglutination inhibition

Line 244-245: Sentence not in English language. Check and translate accordingly.

R/. We agree to the reviewer, we translated the sentence that was in Spanish

Line 249: What is IVD? Define at first mention.

R/. We agree to the reviewer, Influenza D virus was used.

Line 284: What do you mean by “Heq2Neq2”?

R/. We apologize for misunderstanding. “Heq2Neq2” refers to the classical designation to H3N8. See: Tůmová, B., Equine influenza — A segment in influenza virus ecology. Comparative Immunology, Microbiology and Infectious Diseases 1980, 3, (1), 45-59.

Line 321: “qPCR” should read RT-Qpcr

R/. We agree to the reviewer. The sentence was Modified

Figure 3: change “clado” to clade

R/. We agree to the reviewer. The sentence was Modified

Conclusions: this should be expanded to provide in-depth discussion to reflect the vaccine types and vaccination schedule vis-à-vis the differences in the H3N8 viruses in circulation across the Americas, including what has to be done to be ahead of the virus as control measures.

R/. We agree to the reviewer. The conclusion was modified according to the requested.

Reviewer 2 Report

This is a good topic and worthy of a comprehensive review. However, it is very important that the abstract and article are reflected by the title and, given the large body of evidence, that the review covers all aspects of recent research. 

The abstracts should be rewritten to better reflect the title and not merely present general statements from the existing literature about equine influenza globally.

Line 12 & 32 & 62 & 71 & 173 & 226 & 257 & 269 & 273: H7N7 has in the past caused equine influenza but this has not been the case in decades. There have not been epizootics of H7N7 in equines since 1980. Care should be taken not to perpetuate phrases or sentences from older literature. You also cannot state H7N7 is a primary subtype of equine influenza - it was - since it has not been responsible for epizootics in decades. Please clarify lines 257-259: how relevant is this to the point you are trying to make...it is talking about 1981?

Line 13 & 26: Influenza is endemic in the equine population throughout much of the world but not all of the world: it is not present in Australia, Iceland or New Zealand.

Line 18: Influenza vaccination reduces the severity of clinical signs and the amount of virus shed after infection thus reducing the spread of infection to susceptible individuals. This is the case in all species. It is therefore expected that equine influenza could be seen in both vaccinated and unvaccinated horses. It is important to avoid the suggestion that there is somehow lack of vaccine efficacy because vaccinated horses can get signs of equine influenza. Vaccination against equine influenza is key to reducing clinical signs and thus preventing serious disease and even death but also significant impact on high health, high performance horses and significant losses to the equine industry.

Line 22: while influenza viruses can affect different species it is a bit of an overstatement to highlight a potentially zoonotic disease in the abstracts concluding sentence: the review is about the importance and need to vaccinate horses.

Line 82: please provide data to support the stated duration of immunity to influenza infection of 4 months after natural infection. In general horses are likely to have immunity for more than 1 year after infection with equine influenza but booster vaccination may be recommended after 6 months.

Line 95: COPD is more commonly called recurrent airway obstruction and is most commonly associated with allergy to inhaled allergens. Please provide detailed information to support the statement that influenza virus infection increases the risk of RAO in horses.

Lines 156-165: what is the relevance of this to the title of the article?

Line 170: please provide more detail backed by data on the likelihood of fomite transmission and highlight the importance of biosecurity.

Lines 172-185: this is quite difficult to follow: perhaps it would make more sense to review the outbreaks by strain in chronological order, starting from the oldest to the most recent, with the countries in alphabetical order. For example, "H7N7 as last reported COUNTRIES in YEAR. H3N8 was reported in Argentina in 1985 and Uruguay in YEAR. Florida Clade 1 (FC1) was reported in Argentina, Chile and Uruguay in 2011 and 2012."

Lines 186-192: this is a very important paragraph but a very short one in this manuscript. There is a significant body of evidence around the risk in 2-3 year old. It is not physical activity per se that is a risk but the stress put on these individuals in transport, diet changes, mixing, training, etc. Please provide data to support that there is immunosuppression (not immunodepression) rather than insufficient vaccination coverage (lack of adherence to vaccination schedules, mixing of vaccinated and unvaccinated horses, etc). Please provide more support for the final sentence in this paragraph: competing horses may be not be more vulnerable but more exposed because travel and mix with other horses and biosecurity is poor - this is not physical activity per se. Please see also Comparing the effects of non-homogenous mixing patterns on epidemiological outcomes in equine populations: A mathematical modelling study - PubMed (nih.gov) Also, there is no mention of the importance of maternally-derived antibody in foals and the risk of severe disease and death in foals born to unvaccinated or incompletely vaccinated mares.

Lines 193-207: There is more time spent on zoonotic potential while this would generally be considered low. The previous paragraph is more important! Make sure that the terminology used for clinical signs in equines is correct and is not human symptom terminology - serous nasal discharge rather than runny nose, for example. Please read the following article carefully Animal influenza virus infections in humans: A commentary - PubMed (nih.gov) and make sure that the risk of equine influenza transmission to humans is not overstated.

Lines 208-216: Despite lengthy and detailed research and diagnostics, equine influenza has barely been isolated from dogs outside the USA. Two different influenza A viruses have infected and spread among dogs since 2000, and both have been widespread in dogs in North America. The H3N8 canine influenza virus arose in the United States as a variant of equine influenza virus. The H3N2 canine influenza virus arose in Asia (China, Korea) by transfer of an avian influenza virus to dogs. Please provide more evidence for equine influenza being found in dogs in the UK and Australia.

Lines 217-231: this is our most important weapon against equine influenza and warrants clear explanation. Biosecurity and appropriate vaccination are key topics. Please make sure that claims about one equine influenza virus vaccine are adequately supported by published data. Almost all equine influenza virus vaccines (with exception of the live intranasal vaccine) contain adjuvant. This is significantly different from vaccines against human seasonal influenza. Adjuvant plays a clear role and helps to activate both humoral and cell-mediated immunity not only meaning that the duration of immunity of equine influenza vaccines is longer but also (and depending on the vaccine strains) there is potential for protection beyond the vaccine strain. See for example Determining Equine Influenza Virus Vaccine Efficacy-The Specific Contribution of Strain Versus Other Vaccine Attributes - PubMed (nih.gov) The exception is the modified live vector vaccine which includes HA genes of equine influenza and not whole virus or virus subunits. In addition, studies have shown that there is more required than HA to build good immunity in appropriately vaccinated horses with N and other components of the virus likely playing a role. Please mention the requirements of an appropriate vaccination course including revaccination.

Lines 232-293 are very hard to follow and also somehow overlap with lines 172-185. It is fine to present this section by country but I would advise putting the countries in alphabetical order. Please be clear whether there was an outbreak or whether the results are from an epidemiology study. Think of the reader and what their take home message should be rather than writing a long sentence that appear to state (lines 238-242) that the strain isolated in 2018 was from 2012. Lines 244-246 are not in English. Make sure that the section for each country is presented in the same order proposed earlier 1) H7N7 and YEAR 2) H3N8 and years in chronological order. Line 253 transport through Dubai is the standard route for horses around the world - this does not just put Asian horses at risk but risk to other populations. However, this sentence does not belong here as it is about transport and risk. Lines 285-287 mention recent results but the data in the article cited is around 10 years old. Surely there is something more recent from the USA - see for example Frequency of Detection and Prevalence Factors Associated with Common Respiratory Pathogens in Equids with Acute Onset of Fever and/or Respiratory Signs (2008-2021) - PubMed (nih.gov), Investigation of cross-regional spread and evolution of equine influenza H3N8 at US and global scales using Bayesian phylogeography based on balanced subsampling - PubMed (nih.gov), Voluntary surveillance program for equine influenza virus in the United States from 2010 to 2013 - PubMed (nih.gov) and Frequency of shedding of respiratory pathogens in horses recently imported to the United States - PubMed (nih.gov)

Lines 302-303: it is very hard to see the clades you are referring to in the figure. if this is really relevant then consider a focused part of the figure where this is clearly shown

Lines 304-330: I think this should be part of the section Lines 232-293. Lines 312-315 are incorrect: : this does not indicate failure of the vaccination schedule: vaccination does not prevent infection (sterile immunity) but reduces the clinical signs and virus shedding and thus reduces likelihood of spread. This is stated in lines 315-316. Imagine the situation of these horses had not been vaccinated at all or recently!

Line 378: Please make sure you do not misquote or misinterpret reference material: Oladduni et al correctly state "The observation of disease in vaccinated horses may also indicate a possible vaccine breakdown necessitating vaccine verification and appropriate use or administration to forestall future outbreaks." They do not state that there is confirmed vaccination breakdown and in fact in the majority of outbreaks worldwide the majority of cases are in unvaccinated or inappropriately vaccinated horses.

Lines 380-391: this is not entirely correct. Immunity to equine influenza vaccines is complex and is about a lot more than vaccine strain. In addition, some strains have been shown, in combination with an appropriate adjuvant, to provide protection against more than a single clade. Please see please see comments on Lines 217-231 and in particular the article Determining Equine Influenza Virus Vaccine Efficacy-The Specific Contribution of Strain Versus Other Vaccine Attributes - PubMed (nih.gov)

Lines 405-413: you appear to be contradicting your previous statements here by stating that in the US 85% of horses with equine influenza are no vaccinated or have an unknown vaccination status and that only 36% of horses were vaccinated (at all), in other words 64% of the outbreak was in unvaccinated horses. Your point on vaccination coverage is really important but it get lost here: perhaps consider moving this to the Epidemiology section or even better the Prevention and Control section - this is really key!

Line 428 "because the disease is not present"  seems rather odd given that you start by stating this is worldwide and how common it is etc. There are of course other causes of respiratory disease in horses but this statement makes no sense in a review on equine influenza! Why conclude with something that is highly unlikely. Please delete from "or perhaps..." to the end of the sentence.

Lines 430-432 you do not mention the most important measure of all - vaccination - or the importance of biosecurity measures beyond isolation (like regular temperature measurement). You may find these articles useful Report of the International Equine Influenza Roundtable Expert Meeting at Le Touquet, Normandy, February 2013 - PubMed (nih.gov) and Annual booster vaccination and the risk of equine influenza to Thoroughbred racehorses - PubMed (nih.gov)

Abstracts are not suitable reference material for more than around 6-12 months of the date of publication

General

Line 46: please provide more detail on the importance of influenza D to the equine population in the light of the fact that antibodies to influenza D virus have been detected in horses, but no evidence of disease in the field has been reported.

Line 75: equine influenza should be lower case not uppercase

Line 78: serious should read serous

Line 79, 247, 276, 277, 280, 293, 405 and elsewhere: animals get clinical signs not symptoms 

Line 80: please clarify/rewrite this sentence: what do you mean by "spaced out" and define "partial immunity".

Line 80: horses are animals or cases not patients

Line 82. Please start a new sentence with "Morbidity...".

Lines 84-87: this is a very long sentence and would read better split into two or even three sentences. The language on the role of cilia (line 86) requires revision.

Line 249 IVD usually stands for Influenza virus D - is this an error?

Lines 235, 252 delete "vaccinated and unvaccinated"

Line 291 running should read race

Line 354 outbreaks do not spread - delete this word

Line 356 what do you mean by Equestrian? the term equestrian means related to horse riding. do you mean dressage?

Lines 369 -370 Chilean country should read Chile; delete the before "horse movement"

Line 395 insert "a" before "complex issue"

Lines 401-2 delete "those considered"

Lin 414 what is IAV?

Line 418 "in the Americas"

Author Response

RESPONSE TO REVIEWERS COMMENTS - Submission Vaccines-1888753

Please find below our point by point responses to the comments regarding our Manuscript.

The changes are highlighted in Yellow in the file.

We would like to thank the Reviewers and Academic Editor for their helpful suggestions, for critical analysis of the manuscript, and for providing new discussion topics.

Reviewer 2

This is a good topic and worthy of a comprehensive review. However, it is very important that the abstract and article are reflected by the title and, given the large body of evidence, that the review covers all aspects of recent research. 

The abstracts should be rewritten to better reflect the title and not merely present general statements from the existing literature about equine influenza globally.

R/. We agree to the reviewer. The abstract was adjusted to the title, mentioning the equine influenza virus in America, epidemiology, and prevention and control measures, especially vaccination. Vaccination is the best strategy for preventing the infection of new individuals and the presentation of complicated symptoms in those individuals already vaccinated.

Line 12 & 32 & 62 & 71 & 173 & 226 & 257 & 269 & 273: H7N7 has in the past caused equine influenza but this has not been the case in decades. There have not been epizootics of H7N7 in equines since 1980. Care should be taken not to perpetuate phrases or sentences from older literature. You also cannot state H7N7 is a primary subtype of equine influenza - it was - since it has not been responsible for epizootics in decades.

R/. We agree to the reviewer. The information of H7N7 was corrected and modified to present the information as historic reference.

Please clarify lines 257-259: how relevant is this to the point you are trying to make...it is talking about 1981?

R/. We agree to the reviewer. We eliminated the reference to the H7N7 subtype in the outbreaks reported in recent years, however we left information on the history of the virus just as a historic reference of the previous presentation of this virus in the Americas.

Line 13 & 26: Influenza is endemic in the equine population throughout much of the world but not all of the world: it is not present in Australia, Iceland or New Zealand.

R/. We agree to the reviewer. We modified the sentences and included the references.

Line 18: Influenza vaccination reduces the severity of clinical signs and the amount of virus shed after infection thus reducing the spread of infection to susceptible individuals. This is the case in all species. It is therefore expected that equine influenza could be seen in both vaccinated and unvaccinated horses. It is important to avoid the suggestion that there is somehow lack of vaccine efficacy because vaccinated horses can get signs of equine influenza. Vaccination against equine influenza is key to reducing clinical signs and thus preventing serious disease and even death but also significant impact on high health, high performance horses and significant losses to the equine industry.

R/. We totally agree to the reviewer. The information about Vaccine relevance to the simptoms prevention was presented.

Line 22: while influenza viruses can affect different species it is a bit of an overstatement to highlight a potentially zoonotic disease in the abstracts concluding sentence: the review is about the importance and need to vaccinate horses.

R/. We totally agree to the reviewer. Information was deleted from Abstract.

Line 82: please provide data to support the stated duration of immunity to influenza infection of 4 months after natural infection. In general horses are likely to have immunity for more than 1 year after infection with equine influenza but booster vaccination may be recommended after 6 months.

R/. We totally agree to the reviewer. Immunty duration was included in the manuscript.

Line 95: COPD is more commonly called recurrent airway obstruction and is most commonly associated with allergy to inhaled allergens. Please provide detailed information to support the statement that influenza virus infection increases the risk of RAO in horses.

R/. We apologize this misunderstanding of the literature. The sentence was modified and referenced accordingly.

Lines 156-165: what is the relevance of this to the title of the article?

R/. The information was modified to a better presentation of the clínico-patological data of EIV.

Line 170: please provide more detail backed by data on the likelihood of fomite transmission and highlight the importance of biosecurity.

R/. Agree. Text and references were included that indicate the presence of the virus on surfaces and fomites.

Lines 172-185: this is quite difficult to follow: perhaps it would make more sense to review the outbreaks by strain in chronological order, starting from the oldest to the most recent, with the countries in alphabetical order. For example, "H7N7 as last reported COUNTRIES in YEAR. H3N8 was reported in Argentina in 1985 and Uruguay in YEAR. Florida Clade 1 (FC1) was reported in Argentina, Chile and Uruguay in 2011 and 2012."

We agree to the reviewer. We included a Table for a better presentation of the information

Lines 186-192: this is a very important paragraph but a very short one in this manuscript. There is a significant body of evidence around the risk in 2-3 year old. It is not physical activity per se that is a risk but the stress put on these individuals in transport, diet changes, mixing, training, etc. Please provide data to support that there is immunosuppression (not immunodepression) rather than insufficient vaccination coverage (lack of adherence to vaccination schedules, mixing of vaccinated and unvaccinated horses, etc). Please provide more support for the final sentence in this paragraph: competing horses may be not be more vulnerable but more exposed because travel and mix with other horses and biosecurity is poor - this is not physical activity per se. Please see also Comparing the effects of non-homogenous mixing patterns on epidemiological outcomes in equine populations: A mathematical modelling study - PubMed (nih.gov) Also, there is no mention of the importance of maternally-derived antibody in foals and the risk of severe disease and death in foals born to unvaccinated or incompletely vaccinated mares.

R/. We agree to the reviewer. The first sentence was out of context. The paragraph was modified and moved to the prevention and control section. Also in relation to the maternal antibodies and risk of severity of infection, we modified the sentences and included new references.

Lines 193-207: There is more time spent on zoonotic potential while this would generally be considered low. The previous paragraph is more important! Make sure that the terminology used for clinical signs in equines is correct and is not human symptom terminology - serous nasal discharge rather than runny nose, for example. Please read the following article carefully Animal influenza virus infections in humans: A commentary - PubMed (nih.gov) and make sure that the risk of equine influenza transmission to humans is not overstated.

R/. We agree to the reviewer. The information was better presented to avoid overestimation of the zoonotic potential.

Lines 208-216: Despite lengthy and detailed research and diagnostics, equine influenza has barely been isolated from dogs outside the USA. Two different influenza A viruses have infected and spread among dogs since 2000, and both have been widespread in dogs in North America. The H3N8 canine influenza virus arose in the United States as a variant of equine influenza virus. The H3N2 canine influenza virus arose in Asia (China, Korea) by transfer of an avian influenza virus to dogs. Please provide more evidence for equine influenza being found in dogs in the UK and Australia.

R/. We agree to the reviewer. The sentence was modified and recommended information included.

Lines 217-231: this is our most important weapon against equine influenza and warrants clear explanation. Biosecurity and appropriate vaccination are key topics. Please make sure that claims about one equine influenza virus vaccine are adequately supported by published data. Almost all equine influenza virus vaccines (with exception of the live intranasal vaccine) contain adjuvant. This is significantly different from vaccines against human seasonal influenza. Adjuvant plays a clear role and helps to activate both humoral and cell-mediated immunity not only meaning that the duration of immunity of equine influenza vaccines is longer but also (and depending on the vaccine strains) there is potential for protection beyond the vaccine strain. See for example Determining Equine Influenza Virus Vaccine Efficacy-The Specific Contribution of Strain Versus Other Vaccine Attributes - PubMed (nih.gov) The exception is the modified live vector vaccine which includes HA genes of equine influenza and not whole virus or virus subunits. In addition, studies have shown that there is more required than HA to build good immunity in appropriately vaccinated horses with N and other components of the virus likely playing a role. Please mention the requirements of an appropriate vaccination course including revaccination.

R/.We agree to the reviewer. More information about vaccine types, vaccination schemes and adjuvants was included and discussed.

Lines 232-293 are very hard to follow and also somehow overlap with lines 172-185. It is fine to present this section by country but I would advise putting the countries in alphabetical order. Please be clear whether there was an outbreak or whether the results are from an epidemiology study. Think of the reader and what their take home message should be rather than writing a long sentence that appear to state (lines 238-242) that the strain isolated in 2018 was from 2012. Lines 244-246 are not in English. Make sure that the section for each country is presented in the same order proposed earlier 1) H7N7 and YEAR 2) H3N8 and years in chronological order. Line 253 transport through Dubai is the standard route for horses around the world - this does not just put Asian horses at risk but risk to other populations. However, this sentence does not belong here as it is about transport and risk. Lines 285-287 mention recent results but the data in the article cited is around 10 years old. Surely there is something more recent from the USA - see for example Frequency of Detection and Prevalence Factors Associated with Common Respiratory Pathogens in Equids with Acute Onset of Fever and/or Respiratory Signs (2008-2021) - PubMed (nih.gov), Investigation of cross-regional spread and evolution of equine influenza H3N8 at US and global scales using Bayesian phylogeography based on balanced subsampling - PubMed (nih.gov), Voluntary surveillance program for equine influenza virus in the United States from 2010 to 2013 - PubMed (nih.gov) and Frequency of shedding of respiratory pathogens in horses recently imported to the United States - PubMed (nih.gov)

R/ We agree to the reviewer. The name of the section was modified, and changes were made to clarify the type of Study that present the data. Also, the US paragraph was modified to include some of the recommended papers and others recently published.

Lines 302-303: it is very hard to see the clades you are referring to in the figure. if this is really relevant then consider a focused part of the figure where this is clearly shown

R/. Clades were correctly presented in the figure.

Lines 304-330: I think this should be part of the section Lines 232-293. Lines 312-315 are incorrect: : this does not indicate failure of the vaccination schedule: vaccination does not prevent infection (sterile immunity) but reduces the clinical signs and virus shedding and thus reduces likelihood of spread. This is stated in lines 315-316. Imagine the situation of these horses had not been vaccinated at all or recently!

 R/. The suggested adjustment was made. We highligth the importance of mentioning vaccination, and on the prevention of severe symptoms. We also mentioned that the old strains generate good protection due to the slow mutation rate of the virus

Line 378: Please make sure you do not misquote or misinterpret reference material: Oladduni et al correctly state "The observation of disease in vaccinated horses may also indicate a possible vaccine breakdown necessitating vaccine verification and appropriate use or administration to forestall future outbreaks." They do not state that there is confirmed vaccination breakdown and in fact in the majority of outbreaks worldwide the majority of cases are in unvaccinated or inappropriately vaccinated horses.

R/. We agree that Outbreaks have been reported in populations without vaccination or with incorrect and incomplete vaccination for the virus. That information was included

Lines 380-391: this is not entirely correct. Immunity to equine influenza vaccines is complex and is about a lot more than vaccine strain. In addition, some strains have been shown, in combination with an appropriate adjuvant, to provide protection against more than a single clade. Please see please see comments on Lines 217-231 and in particular the article Determining Equine Influenza Virus Vaccine Efficacy-The Specific Contribution of Strain Versus Other Vaccine Attributes - PubMed (nih.gov)

R/. We agree. The information was included. Adjuvant relevance was discussed. Reference was added.

Lines 405-413: you appear to be contradicting your previous statements here by stating that in the US 85% of horses with equine influenza are no vaccinated or have an unknown vaccination status and that only 36% of horses were vaccinated (at all), in other words 64% of the outbreak was in unvaccinated horses. Your point on vaccination coverage is really important but it get lost here: perhaps consider moving this to the Epidemiology section or even better the Prevention and Control section - this is really key!

R/. We agree to the reviewer and apologize for the controversy. The information was modified to highlight the importance of complete vaccination in prevention of disease.

Line 428 "because the disease is not present"  seems rather odd given that you start by stating this is worldwide and how common it is etc. There are of course other causes of respiratory disease in horses but this statement makes no sense in a review on equine influenza! Why conclude with something that is highly unlikely. Please delete from "or perhaps..." to the end of the sentence.

R/. We agree. Answer: it was eliminated the Word “Perhaps”, because we consider that it really is a disease with worldwide distribution and of which there is mandatory notification in most countries of America

Lines 430-432 you do not mention the most important measure of all - vaccination - or the importance of biosecurity measures beyond isolation (like regular temperature measurement). You may find these articles useful Report of the International Equine Influenza Roundtable Expert Meeting at Le Touquet, Normandy, February 2013 - PubMed (nih.gov) and Annual booster vaccination and the risk of equine influenza to Thoroughbred racehorses - PubMed (nih.gov)

R/. We agree to the reviewer. Prevention and control measures for equine influenza were included

Abstracts are not suitable reference material for more than around 6-12 months of the date of publication.

R/. Full abstract was rewritten.

General

Line 46: please provide more detail on the importance of influenza D to the equine population in the light of the fact that antibodies to influenza D virus have been detected in horses, but no evidence of disease in the field has been reported.

R/. influenza D in horses was correctly included and referenced.

Line 75: equine influenza should be lower case not uppercase

R/. Done

Line 78: serious should read serous

R/. Done

Line 79, 247, 276, 277, 280, 293, 405 and elsewhere: animals get clinical signs not symptoms 

R/. Done

Line 80: please clarify/rewrite this sentence: what do you mean by "spaced out" and define "partial immunity".

R/. Done

Line 80: horses are animals or cases not patients

R/. Done

Line 82. Please start a new sentence with "Morbidity...".

R/. Done

Lines 84-87: this is a very long sentence and would read better split into two or even three sentences. The language on the role of cilia (line 86) requires revision.

R/. Done

Line 249 IVD usually stands for Influenza virus D - is this an error?

R/. Done

Lines 235, 252 delete "vaccinated and unvaccinated"

R/. Done

Line 291 running should read race

R/. Done

Line 354 outbreaks do not spread - delete this word

R/. Done

Line 356 what do you mean by Equestrian? the term equestrian means related to horse riding. do you mean dressage?

R/. Done

Lines 369 -370 Chilean country should read Chile; delete the before "horse movement"

R/. Done

Line 395 insert "a" before "complex issue"

R/. Done

Lines 401-2 delete "those considered"

R/. Done

Lin 414 what is IAV?

R/. Done

Line 418 "in the Americas"

R/. Done

Reviewer 3 Report

With the study entitled “Equine Influenza Virus: An old known enemy in the Americas”, the authors presented advance on pathogenesis, epidemiology, prevention and control of EIVs in Americas. It is very interesting for influenza researchers and will guide the prevention and control of equine influenza. However, the manuscript has many little errors and should be improved as follows.

 1. In line 34, please describe clearly that nuclear export protein (NEP) is also referred to as non-structural protein 2, or NS2, because you used the NS2 in the fig.2.

2. Line87, influenza the virus changed into influenza virus.

3. In fig.2 title, Replication changed into replication. In fig.2, remove the wavy line under Golgy and changed it into Golgi.

4. In fig.3, CLADE 2, CLADE 1 but not CLADO2, CLADO1. Green line and blue line need to be defined.

5. In fig.3, please use standard nomenclature for influenza viruses in phylogenetic analysis like A/goose/Guangdong/1/1996 lineage.

6. It is better to add the phylogenetic analysis related with H7N7, in your title is about the review of EIV, or changed your title to limit with H3N8 EIVs.

7. The main mutational rules and the functions of the EIV in recent years should be summarized in table.

8. Please added new reference from 2021 and 2022, I have searched more pater in two years.

9. The format of the references is not uniform.

Author Response

RESPONSE TO REVIEWERS COMMENTS - Submission Vaccines-1888753

Please find below our point by point responses to the comments regarding our Manuscript.

The changes are highlighted in Yellow in the file.

We would like to thank the Reviewers and Academic Editor for their helpful suggestions, for critical analysis of the manuscript, and for providing new discussion topics.

Reviewer 3

With the study entitled “Equine Influenza Virus: An old known enemy in the Americas”, the authors presented advance on pathogenesis, epidemiology, prevention and control of EIVs in Americas. It is very interesting for influenza researchers and will guide the prevention and control of equine influenza. However, the manuscript has many little errors and should be improved as follows.

In line 34, please describe clearly that nuclear export protein (NEP) is also referred to as non-structural protein 2, or NS2, because you used the NS2 in the fig.2.

R/. We completely agree to the reviewer. The sentence was modified..

  1. Line87, influenza the virus changed into influenza virus.

R/. We completely agree to the reviewer. The sentence was modified..

  1. In fig.2 title, Replication changed into replication. In fig.2, remove the wavy line under Golgy and changed it into Golgi.

R/. We completely agree to the reviewer. The sentence was modified..

  1. In fig.3, CLADE 2, CLADE 1 but not CLADO2, CLADO1. Green line and blue line need to be defined.

R/. We completely agree to the reviewer. The Figure was modified.

  1. In fig.3, please use standard nomenclature for influenza viruses in phylogenetic analysis like A/goose/Guangdong/1/1996 lineage.

R/. We completely agree to the reviewer. The Figure was modified.

  1. It is better to add the phylogenetic analysis related with H7N7, in your title is about the review of EIV, or changed your title to limit with H3N8 EIVs.

R/. We completely agree to the reviewer. The Figure description was modified.

  1. The main mutational rules and the functions of the EIV in recent years should be summarized in table.

R/. We agree. A table with this information was added.

  1. Please added new reference from 2021 and 2022, I have searched more pater in two years.

R/. We agree and new references were added.

  1. The format of the references is not uniform.

R/. We apologize for this unintentional typo. We correct the EndNote references that seems to be out of the  right Style.

Round 2

Reviewer 1 Report

All recommended suggestions to the authors were duly implemented in the manuscript except polimerasa in Fig 1 which needs to be changed to polymerase. 

Author Response

Dear reviewer. The figure 1 was corrected.

Reviewer 2 Report

The authors have done a lot of work to respond to the points raised during initial review. They have not however addressed everything and the manuscript suffers from rather poor English. This unfortunately means that some of the critical points about equine influenza are unclear, confusing or even incorrect. For this reason I have stipulated a lot of changes in detail and I would suggest that it be recommended to the authors that they take their time in making sure that their manuscript is complete, without typographical errors and they address the points raised carefully and completely.

Author Response

RESPONSE TO REVIEWERS COMMENTS - Submission Vaccines-1888753

Please find below our point by point responses to the comments regarding our Manuscript. The full manuscript was Edited including changes to grammar, punctuation, phrasing, article use, clarity, style, and flow by the American journal Experts™ (See attached file)

The changes are highlighted in Yellow/Blue in the file.

We would like to thank the Reviewer and Academic Editor for their helpful suggestions, for critical analysis of the manuscript, and for providing new discussion topics.

Reviewer 2

This is a good topic and worthy of a comprehensive review. However, it is very important that the abstract and article are reflected by the title and, given the large body of evidence, that the review covers all aspects of recent research.

  1. The abstracts should be rewritten to better reflect the title and not merely present general statements from the existing literature about equine influenza globally.

R/. We agree to the reviewer. The abstract was adjusted to the title, mentioning the equine influenza virus in America, epidemiology, and prevention and control measures, especially vaccination. Vaccination is the best strategy for preventing the infection of new individuals and the presentation of complicated symptoms in those individuals already vaccinated. Lines 22- 25

  1. Line 12 & 32 & 62 & 71 & 173 & 226 & 257 & 269 & 273: H7N7 has in the past caused equine influenza but this has not been the case in decades. There have not been epizootics of H7N7 in equines since 1980. Care should be taken not to perpetuate phrases or sentences from older literature. You also cannot state H7N7 is a primary subtype of equine influenza - it was - since it has not been responsible for epizootics in decades.

R/. We agree to the reviewer. The information of H7N7 was corrected and modified to present the information as historic reference. A historic mention to H7N7 was made in lines 40 and 77.

  1. Please clarify lines 257-259: how relevant is this to the point you are trying to make...it is talking about 1981,

R/. We agree to the reviewer. We eliminated the reference to the H7N7 subtype in the outbreaks reported in recent years, however we left information on the history of the virus just as a historic reference of the previous presentation of this virus in the America. Lines 40 and 77.

  1. Line 13 & 26: Influenza is endemic in the equine population throughout much of the world but not all of the world: it is not present in Australia, Iceland or New Zealand in the line 31

R/. We agree to the reviewer. We modified the sentences and included the references. Information was added in line 31

  1. Line 18: Influenza vaccination reduces the severity of clinical signs and the amount of virus shed after infection thus reducing the spread of infection to susceptible individuals. This is the case in all species. It is therefore expected that equine influenza could be seen in both vaccinated and unvaccinated horses. It is important to avoid the suggestion that there is somehow lack of vaccine efficacy because vaccinated horses can get signs of equine influenza. Vaccination against equine influenza is key to reducing clinical signs and thus preventing serious disease and even death but also significant impact on high health, high performance horses and significant losses to the equine industry.

R/. We totally agree to the reviewer. The information about Vaccine relevance to the symptom’s prevention was presented. Information was added in lines 495-498 We also added references from Paillot 2012 y Dionisio 2021

  1. Line 22: while influenza viruses can affect different species it is a bit of an overstatement to highlight a potentially zoonotic disease in the abstracts concluding sentence: the review is about the importance and need to vaccinate horses.

R/. We totally agree to the reviewer. Information was deleted from Abstract.

  1. Line 82: please provide data to support the stated duration of immunity to influenza infection of 4 months after natural infection. In general horses are likely to have immunity for more than 1 year after infection with equine influenza but booster vaccination may be recommended after 6 months.

R/. We totally agree to the reviewer. Immunity duration was included in the manuscript. Line 150 and 511

  1. Line 95: COPD is more commonly called recurrent airway obstruction and is most commonly associated with allergy to inhaled allergens. Please provide detailed information to support the statement that influenza virus infection increases the risk of RAO in horses.

R/. We apologize this misunderstanding of the literature. The sentence was modified and referenced accordingly. COPD information was deleted. We also included physiopathology information. Lines 157-160

  1. Lines 156-165: what is the relevance of this to the title of the article?

R/. The information was modified to a better presentation of the clinicopathological data of EIV. Lines Lines 157-160 and we delete EIV animal models

  1. Line 170: please provide more detail backed by data on the likelihood of fomite transmission and highlight the importance of biosecurity.

R/. Agree. Text and references were included that indicate the presence of the virus on surfaces and fomites, line 181 -182 In lines 512- 513 we included new information about fomites and its role in prevention and control

  1. Lines 172-185: this is quite difficult to follow: perhaps it would make more sense to review the outbreaks by strain in chronological order, starting from the oldest to the most recent, with the countries in alphabetical order. For example, "H7N7 as last reported COUNTRIES in YEAR. H3N8 was reported in Argentina in 1985 and Uruguay in YEAR. Florida Clade 1 (FC1) was reported in Argentina, Chile and Uruguay in 2011 and 2012."

R/. We agree to the reviewer. We included a Table 1 for a better presentation of the information

  1. Lines 186-192: this is a very important paragraph but a very short one in this manuscript. There is a significant body of evidence around the risk in 2-3 year old. It is not physical activity per se that is a risk but the stress put on these individuals in transport, diet changes, mixing, training, etc. Please provide data to support that there is immunosuppression (not immunodepression) rather than insufficient vaccination coverage (lack of adherence to vaccination schedules, mixing of vaccinated and unvaccinated horses, etc). Please provide more support for the final sentence in this paragraph: competing horses may be not be more vulnerable but more exposed because travel and mix with other horses and biosecurity is poor - this is not physical activity per se. Please see also Comparing the effects of non-homogenous mixing patterns on epidemiological outcomes in equine populations: A mathematical modelling study - PubMed (nih.gov) Also, there is no mention of the importance of maternally-derived antibody in foals and the risk of severe disease and death in foals born to unvaccinated or incompletely vaccinated mares.

R/. We agree to the reviewer. The first sentence was out of context. The paragraph was modified and moved to the prevention and control section. Also in relation to the maternal antibodies and risk of severity of infection, we modified the sentences and included new references about the risk in 2-3 years. Lines 455-471 and  280-283 about maternally antibodies. Also we include the suggested references

  1. Lines 193-207: There is more time spent on zoonotic potential while this would generally be considered low. The previous paragraph is more important! Make sure that the terminology used for clinical signs in equines is correct and is not human symptom terminology - serous nasal discharge rather than runny nose, for example. Please read the following article carefully Animal influenza virus infections in humans: A commentary - PubMed (nih.gov) and make sure that the risk of equine influenza transmission to humans is not overstated.

R/. We agree to the reviewer. The information was better presented to avoid overestimation of the zoonotic potential.

  1. Lines 208-216: Despite lengthy and detailed research and diagnostics, equine influenza has barely been isolated from dogs outside the USA. Two different influenza A viruses have infected and spread among dogs since 2000, and both have been widespread in dogs in North America. The H3N8 canine influenza virus arose in the United States as a variant of equine influenza virus. The H3N2 canine influenza virus arose in Asia (China, Korea) by transfer of an avian influenza virus to dogs. Please provide more evidence for equine influenza being found in dogs in the UK and Australia.

R/. We agree to the reviewer. The sentence was modified and recommended information included in Lines 236 to 242

  1. Lines 217-231: this is our most important weapon against equine influenza and warrants clear explanation. Biosecurity and appropriate vaccination are key topics. Please make sure that claims about one equine influenza virus vaccine are adequately supported by published data. Almost all equine influenza virus vaccines (with exception of the live intranasal vaccine) contain adjuvant. This is significantly different from vaccines against human seasonal influenza. Adjuvant plays a clear role and helps to activate both humoral and cell-mediated immunity not only meaning that the duration of immunity of equine influenza vaccines is longer but also (and depending on the vaccine strains) there is potential for protection beyond the vaccine strain. See for example Determining Equine Influenza Virus Vaccine Efficacy-The Specific Contribution of Strain Versus Other Vaccine Attributes - PubMed (nih.gov) The exception is the modified live vector vaccine which includes HA genes of equine influenza and not whole virus or virus subunits. In addition, studies have shown that there is more required than HA to build good immunity in appropriately vaccinated horses with N and other components of the virus likely playing a role. Please mention the requirements of an appropriate vaccination course including revaccination.

R/. We agree to the reviewer. More information about vaccine types, vaccination schemes and adjuvants was included and discussed. The information related to Vaccines and adjuvants was added in lines 493-497  and lines 328-336. Biosafety related information’s was included in lines 485- 488 and about vaccination plan 499-505.

  1. Lines 232-293 are very hard to follow and also somehow overlap with lines 172-185. It is fine to present this section by country but I would advise putting the countries in alphabetical order. Please be clear whether there was an outbreak or whether the results are from an epidemiology study. Think of the reader and what their take home message should be rather than writing a long sentence that appear to state (lines 238-242) that the strain isolated in 2018 was from 2012. Lines 244-246 are not in English. Make sure that the section for each country is presented in the same order proposed earlier 1) H7N7 and YEAR 2) H3N8 and years in chronological order. Line 253 transport through Dubai is the standard route for horses around the world - this does not just put Asian horses at risk but risk to other populations. However, this sentence does not belong here as it is about transport and risk. Lines 285-287 mention recent results but the data in the article cited is around 10 years old. Surely there is something more recent from the USA - see for example Frequency of Detection and Prevalence Factors Associated with Common Respiratory Pathogens in Equids with Acute Onset of Fever and/or Respiratory Signs (2008-2021) - PubMed (nih.gov), Investigation of cross-regional spread and evolution of equine influenza H3N8 at US and global scales using Bayesian phylogeography based on balanced subsampling - PubMed (nih.gov), Voluntary surveillance program for equine influenza virus in the United States from 2010 to 2013 - PubMed (nih.gov) and Frequency of shedding of respiratory pathogens in horses recently imported to the United States - PubMed (nih.gov)

R/ We agree to the reviewer. The name of the section was modified, and changes were made to clarify the type of Study that present the data. Also, the US paragraph was modified to include some of the recommended papers and others recently published, and the information is in alphabetical order, US EIV information was updated in Lines 413-419

  1. Lines 302-303: it is very hard to see the clades you are referring to in the figure. if this is really relevant then consider a focused part of the figure where this is clearly shown

R/. Clades were correctly presented in the figure 3

  1. Lines 304-330: I think this should be part of the section Lines 232-293. Lines 312-315 are incorrect: : this does not indicate failure of the vaccination schedule: vaccination does not prevent infection (sterile immunity) but reduces the clinical signs and virus shedding and thus reduces likelihood of spread. This is stated in lines 315-316. Imagine the situation of these horses had not been vaccinated at all or recently!

R/. The suggested adjustment was made.  The 2018 LatinAmerican outbreak information was included in Lines  439 to 441.

We highlight the importance of vaccination on sings reduction and prevention - Lines 439-441 and 357 We also mentioned that the old strains generate good protection due to the slow mutation rate of the virus--- Line  329-331

  1. Line 378: Please make sure you do not misquote or misinterpret reference material: Oladduni et al correctly state "The observation of disease in vaccinated horses may also indicate a possible vaccine breakdown necessitating vaccine verification and appropriate use or administration to forestall future outbreaks." They do not state that there is confirmed vaccination breakdown and in fact in the majority of outbreaks worldwide the majority of cases are in unvaccinated or inappropriately vaccinated horses.

R/. We agree that Outbreaks have been reported in populations without vaccination or with incorrect and incomplete vaccination for the virus. That information was included in lines 315 to 318

  1. Lines 380-391: this is not entirely correct. Immunity to equine influenza vaccines is complex and is about a lot more than vaccine strain. In addition, some strains have been shown, in combination with an appropriate adjuvant, to provide protection against more than a single clade. Please see please see comments on Lines 217-231 and in particular the article Determining Equine Influenza Virus Vaccine Efficacy-The Specific Contribution of Strain Versus Other Vaccine Attributes - PubMed (nih.gov)

R/. We agree. The information was included. Adjuvant relevance was discussed. Reference was added. The information was included in line  483-486

  1. Lines 405-413: you appear to be contradicting your previous statements here by stating that in the US 85% of horses with equine influenza are no vaccinated or have an unknown vaccination status and that only 36% of horses were vaccinated (at all), in other words 64% of the outbreak was in unvaccinated horses. Your point on vaccination coverage is really important but it get lost here: perhaps consider moving this to the Epidemiology section or even better the Prevention and Control section - this is really key!

R/. We agree to the reviewer and apologize for the controversy. The information was modified to highlight the importance of complete vaccination in prevention of disease. Lines  354 a 358

  1. Line 428 "because the disease is not present" seems rather odd given that you start by stating this is worldwide and how common it is etc. There are of course other causes of respiratory disease in horses but this statement makes no sense in a review on equine influenza! Why conclude with something that is highly unlikely. Please delete from "or perhaps..." to the end of the sentence.

R/. We agree. because we consider that it really is a disease with worldwide distribution and of which there is mandatory notification in most countries of America.

Lines 430-432 you do not mention the most important measure of all - vaccination - or the importance of biosecurity measures beyond isolation (like regular temperature measurement). You may find these articles useful Report of the International Equine Influenza Roundtable Expert Meeting at Le Touquet, Normandy, February 2013 - PubMed (nih.gov) and Annual booster vaccination and the risk of equine influenza to Thoroughbred racehorses - PubMed (nih.gov)

R/. We agree to the reviewer. Prevention and control measures for equine influenza were incorporated. We also included the recommended references. Lines  511-519.

  1. Abstracts are not suitable reference material for more than around 6-12 months of the date of publication.

R/. Full abstract was rewritten. lines 13- 24

General

  1. Line 46: please provide more detail on the importance of influenza D to the equine population in the light of the fact that antibodies to influenza D virus have been detected in horses, but no evidence of disease in the field has been reported.

R/. influenza D in horses was correctly included and referenced. Lines  53 to 56

  1. Line 75: equine influenza should be lower case not uppercase

R/. Done in line 83

  1. Line 78: serious should read serous

R/. Done line 147

  1. Line 79, 247, 276, 277, 280, 293, 405 and elsewhere: animals get clinical signs not symptoms

R/. Done. Lines 77, 132,148,283,349,206,383,385,390,402,403, 412,437,438,486

  1. Line 80: please clarify/rewrite this sentence: what do you mean by "spaced out" and define "partial immunity".

R/. Done, it means that it isn’t  present in vaccinated animals, lines148-149, we eliminated the term partial inmmunity, because does not has sense with the text

  1. Line 80: horses are animals or cases not patients

R/. Done, we changed patients for cases 159, 160,187,231,328, 406,442,444.

  1. Line 82. Please start a new sentence with " Morbidity...".

R/. Done, The sentence beguin with: morbidity in line 151

  1. Lines 84-87: this is a very long sentence and would read better split into two or even three sentences. The language on the role of cilia (line 86) requires revision.

R/. Done, we divided the sentences, and corrected the language  lines 89-90

  1. Line 249 IVD usually stands for Influenza virus D - is this an error?

R/. Done, sorry for the mistake we changed in line 420

  1. Lines 235, 252 delete "vaccinated and unvaccinated"

R/. Done, we eliminated "vaccinated and unvaccinated" and changed by horses

  1. Line 291 running should read race

R/. Done line 241, we changed running by race in line 383

  1. Line 354 outbreaks do not spread - delete this word

R/. Done, we chaged by passed in line 302

  1. Line 356 what do you mean by Equestrian? the term equestrian means related to horse riding. do you mean dressage?

R/. Done, it means riding horse,

  1. Lines 369 -370 Chilean country should read Chile; delete the before "horse movement"

R/. Done, horse ridding in line  288

  1. Line 395 insert "a" before "complex issue"

R/. Done we include a before complex issue 339

  1. Lines 401-2 delete "those considered"

R/. Done we change by especially in line 345

  1. Lin 414 what is IAV?

R/. Done, IAV means ( Influenza A virus) line 359

  1. Line 418 "in the Americas"

R/. Done, we include in the Americas in lines 521-522

Reviewer 3 Report

Please recheck the references and unify the format of paper titles and journal titles. For example, the journal title of article 53 references and article 94 references can be changed to “Plos One”. However, there are two different formats (“PLoS One” and “PLOS ONE”) in the manuscript. And you can manually correct similar errors.

Author Response

Dear Reviewer. Tanks for your sugestion. Endnote references were updated according to the Journal format, incluiding misspelling.

Round 3

Reviewer 2 Report

The authors are to be congratulated on their hard work. They have considerably improved the manuscript, and this will undoubtedly benefit readers of it.